# Context-Aware Replanning with Pre-explored Semantic Map for Object Navigation

**Po-Chen Ko**[1,*]    **Hung-Ting Su**[1,*]    **Ching-Yuan Chen**[1,*]    **Jia-Fong Yeh**[1]
**Min Sun**[2]    **Winston H. Hsu**[1,3]
*Equal Contribution
[1]National Taiwan University    [2]National Tsing Hua University
[3]Mobile Drive Technology

**Abstract:** Pre-explored Semantic Maps, constructed through prior exploration using visual language models (VLMs), have proven effective as foundational elements for training-free robotic applications. However, existing approaches assume the map's accuracy and do not provide effective mechanisms for revising decisions based on incorrect maps. To address this, we introduce **Context-Aware Replanning (CARe)**, which estimates map uncertainty through confidence scores and multi-view consistency, enabling the agent to revise erroneous decisions stemming from inaccurate maps without requiring additional labels. We demonstrate the effectiveness of our proposed method by integrating it with two modern mapping backbones, VLMaps and OpenMask3D, and observe significant performance improvements in object navigation tasks. More details can be found on the project page: `https://care-maps.github.io/`.

**Keywords:** Object Navigation, Vision-Language Models, Uncertainty Measurement, Training-Free Replanning

## 1  Introduction

Navigating in indoor environments to locate and reach target objects is a fundamental capability for robotic applications. Conventional training-based object navigation methods necessitate extensive annotations, meticulous model design, and prolonged training periods to effectively align the control actions with visual perception. Advancements in visual language models (VLMs) have led to the development of modular approaches that separate perception from actions, utilizing pre-trained knowledge. Under this framework, visual perception can be independently learned without direct control, making exploration prior to task execution an effective strategy. Pre-explored Semantic Map [1, 2, 3, 4, 5], constructed through prior exploration and using visual language models (VLMs), has become a fundamental backbone for robotics tasks. By constructing the map during environmental exploration, Pre-explored Semantic Map equips the agent with pre-existing knowledge of the environment, facilitating training-free robotic tasks such as manipulation motion planning [6], interactive exploration [7], and zero-shot object navigation [5]. However, current approaches presume that Pre-explored Semantic Map is always accurate—unbiased and noise-free—and they lack effective mechanisms for revising decisions based on incorrect maps. This assumption is flawed for real-world applications, where visual perception must contend with diverse environments, including varying lighting conditions, textures, weather, and dynamic elements. Consequently, visual perception cannot be assumed to be perfect. In addition, evaluating the quality of the Pre-explored Semantic Map is also challenging, as there are no existing labels for comparison.

Intuitively, the map-based agent plans its path by retrieving the goal from the map, where retrieval is facilitated by sorting confidence scores based on map-query matching. Therefore, should the initial attempt fail, the agent can proceed to another unvisited location with the highest confidence score. Nevertheless, this approach assumes the map is accurate. However, failure to retrieve the map

8th Conference on Robot Learning (CoRL 2024), Munich, Germany.

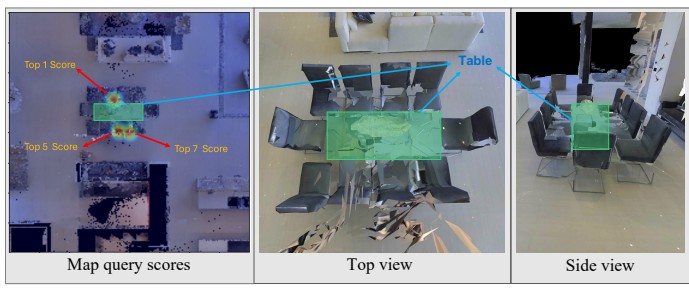

Map query scores | Top view | Side view

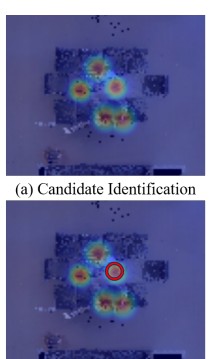

(a) Candidate Identification

(b) Selection w/ Uncertainty

Figure 1: **Motivation.** When the initial map query fails, high-confidence regions also tend to fail due to biases in visual perception. (Query: Table, Grounding: Chair)

Figure 2: **Method Overview.**

with the highest confidence score may indicate a mismatch or error in the map-query alignment. Given the observation that the visual observation upon which the map is based fails to satisfy the query, we hypothesize that integrating **Uncertainty** measurement into the decision-making process can enhance the agent's ability to cope with inaccuracies in the map. By prioritizing areas with the greatest uncertainty, the agent targets high-uncertainty points for maximum information gain. This improves performance because initial prediction failures often result from biases or errors in the map data. Replanning based solely on confidence scores can perpetuate these biases. High-uncertainty areas, however, are less influenced by existing biases. Therefore, integrating uncertainty measurement reduces bias impact and increases the likelihood of successfully finding the goal. In addition, the same region can be observed from multiple viewpoints. If the VLM predictions from these multiple viewpoints are not aligned, the prediction is likely incorrect. Therefore, **Multi-view Consistency** can also serve as an unsupervised metric for agent replanning.

To this end, we propose a novel **Context-Aware Replanning (CARe)** to revise the plan when the initial planning fails. Specifically, we first select $k$ candidate regions as a strategy to mitigate the limitations of imperfect perception systems. While the top-ranked region might not always be reliable due to potential inaccuracies in perception, selecting a broader set of top candidates ensures a more robust assessment. Afterward, we measure uncertainty and multi-view consistency for candidate regions and select the best candidate based on the measured values. Our evaluation in two popular Pre-explored Semantic Map backbones, VLMaps [5] and OpenMask3D [3], showcased that our method consistently outperforms the strategy of selecting the highest-scored unvisited region.

The contribution of this work is summarized as follows:

- We propose a novel Context-Aware Replanning (CARe), which replans based on the context of an incorrect Pre-explored Semantic Map when the initial task fails.
- We design two variants of Context-Aware Replanning, based on uncertainty and multi-view inconsistency, without requiring additional annotations.
- Our method consistently outperforms the strategy of selecting the highest-scored unvisited region using two different backbones [5, 3], demonstrating its effectiveness and robustness.

## 2 Related Work

Visual Language Models (VLMs) serve as the backbone of modern modular training-free robotic applications by aligning visual perception with language, enabling robots to understand and follow natural language instructions. CLIP [8] achieves this alignment through contrastive learning, forming the foundation for various training-free robotic applications. For instance, OpenMask3D [3] and CLIP-Fields [2] utilize CLIP to construct maps. Additionally, several open-vocabulary visual perception tools, such as GLIP [9] for object detection and LSeg [10] for semantic segmentation, are crucial for robotic applications. For instance, VLMaps [5] uses LSeg to build its maps. While VLMs achieve significant performance on various computer vision tasks, robotic applications must

handle diverse environments with varying lighting conditions, textures, weather, and object categories. Therefore, accounting for and replanning with errors in maps is crucial when leveraging maps constructed with VLMs.

# 3   Method

In this section, we first introduce the background of our method in 3.1. Next, we illustrate the motivating concept behind our method to address a significant issue in current approaches in Section 3.2. We then detail our approach, which primarily focuses on leveraging uncertainty measures to select from a set of high-confidence candidates, as described in Section 3.3. Within this section, we further explain the single and multi-view uncertainty measures that our method utilizes. Subsequently, we outline the strategies we tested to generate the set of high-confidence candidates in Section 3.4. Finally, we describe the maps used in our experiments and discuss how they were adapted to integrate with our method in Section 3.5. Figure 2 illustrates a brief overview of our method.

## 3.1   Map-Based Navigation.

### 3.1.1   Goal-Oriented Object Navigation as Object Retrieval from Map.

Navigating in a pre-explored environment typically involves an agent's internal representation of the environment, referred to as the map. Recent methods [3, 5] have reframed the goal-oriented object navigation task into an object retrieval task on the map by employing a path planner that devises a route from the agent's current position to the target, coupled with a policy that actuates the agent to follow this planned path.

### 3.1.2   Replanning on Failure.

In practical scenarios, retrieval from the map is not always successful. In such cases, the capability to automatically replan becomes crucial for an autonomous agent, as it circumvents the need for costly human intervention. An intuitive approach to replanning would involve selecting the unvisited candidate location with the highest retrieval score.

## 3.2   Uncertainty aware object retrieval.

The conventional way to use navigation maps typically involves using a text query to retrieve the candidate with the highest matching score. Such a score is calculated with pretrained visual-language grounding models such as CLIP[8]. These grounding models have demonstrated impressive generalization abilities due to the large dataset used in the pertaining phase, which is probably why most of the works navigating with maps assume that the map is perfect. Nevertheless, we argue that despite the size of the training dataset, there might still be bias introduced by the model architecture or the dataset, which causes the user expectation of a query to diverge from the model's belief. Furthermore, during the construction phase of the map, the model prediction might be interfered with by surrounding objects or some view-dependent bias, causing the extracted feature to be noisy.

To address these issues, our framework integrates uncertainty measures to enhance retrieval accuracy. These measures can potentially reflect biases or noise associated with a candidate, thereby improving retrieval performance.

## 3.3   Uncertainty Measures.

### 3.3.1   Single-View Uncertainty Measures.

We hypothesize that when the grounding model retrieves the wrong object with high confidence, the model might have shown some bias in the object category. This is a common failure mode in conventional map-based navigation because the algorithm always chooses the candidate with the highest score/confidence. When the model bias occurs, solely looking at the confidence/score might give misleading retrieval results. On the other hand, completely ignoring the confidence/score might also be suboptimal, as it disregards valuable information from the strong grounding model. In

this case, encouraging the retrieval of high confidence and high classification uncertainty candidates might be helpful as it both leverages the confidence information and mitigates the impact of potential biases by encouraging the exploration of uncertain candidates. Specifically, we first identify a list of high-confidence candidates and choose the candidate in the list with the highest entropy.

**3.3.1.1 Entropy:** Let $C = \{c_1, c_2, \ldots, c_n\}$ be the set of high-confidence candidates. For each $c_i$, we compute the entropy $H(c_i)$ of its classification distribution $P_i = \{p_1, p_2, \ldots, p_k\}$ as:

$$H(c_i) = -\sum_{j=1}^{k} p_j \log p_j$$

where $p_j$ is the probability of candidate $c_i$ belonging to class $j$. We select the candidate $c^*$ with the highest entropy:

$$c^* = \arg\max_{c_i \in C_{\text{high}}} H(c_i)$$

With this definition, a high entropy indicates that the classification distribution resembles a spread-out probability distribution, suggesting that the model considers several classes as plausible alternatives, thereby reflecting uncertainty. This approach ensures that we are not only considering candidates with high confidence but those with high uncertainty, thereby reducing the likelihood of biased retrievals. For the specific method of selecting high-confidence candidates, please refer to Section 3.4.

### 3.3.2 Multi-View Uncertainty Measures.

When the feature of a candidate is extracted by a small number of views, the resulting feature can easily be interfered by nearby objects or view-dependent model bias. To mitigate this issue, we propose to encourage multi-view consistency when retrieving the candidate. Similar to the single-view case, we first identify a list of high-confidence candidates and then encourage the retrieval of low-multi-view inconsistency (uncertainty) candidates. We experimented with two kinds of multi-view uncertainty measures: channel-average feature standard error and mean pairwise KL divergence on multi-view classification probabilities.

Let $C = \{c_1, c_2, \ldots, c_n\}$ be the set of candidates, and $V_i = \{v_1, v_2, \ldots, v_{m_i}\}$ be the set of views for candidate $c_i$. For a candidate $c_i$ with only a single view feature, we set the uncertainty score $U(c_i)$ to infinity.

**3.3.2.1 Channel-average Feature Standard Error:** For each candidate $c_i$ with multiple views, the standard error $SE(c_i)$ is computed as:

$$SE(c_i) = \frac{1}{d}\sum_{k=1}^{d} \frac{\sigma_k}{\sqrt{m_i}}$$

where $\sigma_k$ is the standard deviation of the $k$-th feature across all views, $d$ is the dimensionality of the feature vector, and $m_i$ is the number of views for candidate $c_i$.

Finally, we choose the candidate $c^*$ with the lowest standard error. Since most maps utilize the mean feature from multiple views for retrieval, a low standard error on these multi-view features indicates that the mean feature is statistically more reliable. Therefore encouraging low standard error could potentially reduce noisy retrievals.

**3.3.2.2 Mean Pairwise KL Divergence:** For each candidate $c_i$, let $P_{ij} = \{p_{ij1}, p_{ij2}, \ldots, p_{ijk}\}$ be the classification probability distribution for view $v_j$. The mean pairwise KL divergence $D_{KL}(c_i)$ is computed as:

$$D_{KL}(c_i) = \frac{2}{m_i(m_i - 1)} \sum_{j=1}^{m_i-1} \sum_{l=j+1}^{m_i} \left( \frac{1}{2} D_{KL}(P_{ij} \parallel P_{il}) + \frac{1}{2} D_{KL}(P_{il} \parallel P_{ij}) \right)$$

where $D_{KL}(P \parallel Q)$ is the KL divergence between two probability distributions $P$ and $Q$.

We then choose the candidate $c^*$ with the lowest mean pairwise KL divergence. A low mean pairwise KL divergence indicates that the classification distribution is consistent across multiple views. Similar to the standard error, promoting consistency in the classification distribution could help reduce noisy retrievals.

## 3.4 Selection of High Confidence Candidates.

In perception process, the highest-ranked class may not always be accurate because of potential errors or limitations in the system's capabilities. By selecting the top $k$ candidate instead of relying solely on the top-ranked one, we mitigate the risk of inaccuracies and enhance the robustness of the assessment. This approach allows us to account for possible errors and improve the overall reliability by considering multiple promising candidates.

### 3.4.1 Top-k Confidence:

Following the conventional and intuitive method, we simply choose the point with the highest confidence:

$$\arg \max_i \text{conf}_i^{\text{cls}}$$

where $\text{conf}_i^{\text{cls}}$ is the confidence, which is the probability or LSeg[10] score in the case of OpenMask3D[11] and VLMaps[5] respectively, of a single point $i$ with the specified goal class $cls$ in the task.

### 3.4.2 Top-k Category:

Similar to the concept of top-k accuracy (Acc@K), we filter the point by whether its top-k confident predicted class contains the specified goal class or not, which can be formulated as follows:

$$\text{Filter}(cls, \text{conf}_i, k) = \begin{cases} \text{True,} & \text{if } cls \in (\text{argsort}_{cls} \text{ conf}_i) [0:k] \\ \text{False,} & \text{otherwise} \end{cases}$$

where $cls$ is the target class and $(\text{argsort}_{cls} \text{ conf}_i) [0:k]$ is the highest $k$ class prediction for the point $i$.

## 3.5 Maps.

We experimented with the proposed uncertainty-aware navigation method on two popular maps for object navigation: OpenMask3D [11] and VLMaps [5]. As described in Section 3.1.1, we focus on retrieval from the map by assuming successful path planning and following in OpenMask3D. In VLMaps, we employed the built-in path planner and follower from the HabitatSim simulator [12, 13, 14].

### 3.5.1 OpenMask3D.

OpenMask3D[11] takes in the scene point cloud and posed RGB images, generates class-agnostic 3D masks on the point cloud, and uses the posed RGB images where the object is visible to provide semantic features. The features are calculated with CLIP with cropped images of the corresponding object and thus enable retrieval later on. If the object is visible in multiple views, OpenMask3D takes the average of multi-view features. The implementation details are further described in A.2.

### 3.5.2 VLMaps.

VLMaps takes the RGB-D image and its corresponding pose as input. It first generate the local point cloud which is then projected to the world coordinate frame and the map position with the information of camera poses. After that, the RGB image is fed into Visual Encoder of LSeg[10] to get its image feature, which will then be projected to the corresponding map position. Though we do not modify the process of building the map, we store some more data such as the standard error and KL divergence of each point on the map to decrease the size of the map and avoid repetitive

| Replan Strategy | Selection Criteria | k=2 | k=4 | k=8 | k=16 | k=40 |
|---|---|---|---|---|---|---|
| No replan (top1 acc) | - | 12.09 | 12.09 | 12.09 | 12.09 | 12.09 |
| Oracle | - | 36.40 | 36.40 | 36.40 | 36.40 | 36.40 |
| Max confidence (top2 acc) | - | 17.05 | 17.05 | 17.05 | 17.05 | 17.05 |
| Random replan | - | 13.77 | 13.77 | 13.77 | 13.77 | 13.77 |
| Random from top-k | confidence (3.4.1) | *17.12* | 16.92 | 16.81 | 16.34 | 14.97 |
| | category (3.4.2) | 13.79 | 13.72 | 13.54 | 13.63 | 13.44 |
| Max entropy (3.3.1) | confidence | 17.49 | 18.11 | 18.20 | 17.66 | 15.20 |
| | category | 17.77 | 17.71 | 17.78 | 17.69 | 17.09 |
| Min stderr (3.3.2.1) | confidence | 17.64 | 17.71 | 17.65 | 16.40 | 15.19 |
| | category | 18.07 | 18.00 | 18.00 | 17.86 | 18.07 |
| Min pwKL (3.3.2.2) | confidence | 17.49 | 17.36 | 17.73 | 17.44 | 16.07 |
| | category | **18.09** | **18.13** | **18.29** | **18.33** | **18.75** |

Table 1: **OM3D Object Retrieval Success Rates.** The upper half of the table presents the baseline methods, as described in 4.1.1, while the lower half displays variants of our method. The 'Replan Strategy' column indicates the replanning strategies used, as detailed in 3.3. The 'Selection Criteria' column specifies the criteria employed to generate the high-confidence candidate set, as described in 3.4. The best-performing entry for each column, excluding the Oracle, is **bolded** to emphasize the best-performing method given $k$. The best-performing entry for each row is highlighted to showcase the performance of each method when a hyper-parameter search on $k$ is available. Finally, the best-performing baseline entry is *italicized* for easier comparison.

computation in the evaluation process. When calculating the Channel-average Feature Standard Error, we incorporate the distance weighting [1, 5] information of each feature which is utilized when building the map. For more details, please refer to A.3.

# 4 Experiments

## 4.1 OpenMask3D Object Retrieval Benchmark.

To validate our method, we conducted an experiment in the Matterport3D environment [13, 14, 12] using OpenMask3D on a two-shot object retrieval task. In this task, a second retrieval attempt is allowed if the first one fails. Each room is benchmarked separately, and the results are presented in Table 1. Note that in all baselines, the first candidate is always retrieved based on maximum confidence in the query class.

We utilized the Matterport raw_category, which contains 1658 classes, as the vocabulary required by OpenMask3D. In total, 10 scans (houses), 214 rooms, and 5370 object instances were evaluated in this experiment.

### 4.1.1 Baselines.

- **No replan:** This is the top-1 retrieval accuracy without a second retrieval attempt.
- **Oracle:** This is the upper-bound baseline where we count it success if any of the predicted 3D masks, regardless of its class scores, matches the GT one. This can also be seen as a top-infinity accuracy baseline.
- **Max confidence (top2 acc):** This is a naive baseline where the candidate with the second-highest confidence is selected when the first one fails, which matches the typical usage of navigation maps.
- **Random replan:** This baseline randomly selects a candidate from all unvisited candidates as the second attempt.
- **Random from top-k:** This baseline randomly selects a candidate from the high-confidence candidates set with the corresponding selection criteria as described in 3.4.

| Replan Strategy | Selection Criteria | k=4 | k=8 | k=16 | k=40 | k=100 |
|---|---|---|---|---|---|---|
| No replan | - | 54.4 | 54.4 | 54.4 | 54.4 | 54.4 |
| Max confidence | - | *67.0* | *67.0* | *67.0* | *67.0* | *67.0* |
| Random replan | - | 56.2 | 56.2 | 56.2 | 56.2 | 56.2 |
| Max entropy (3.3.1) | confidence (3.4.1) | 81.0 | 67.6 | 67.9 | 66.8 | 65.4 |
| | category (3.4.2) | 70.4 | 67.3 | 64.2 | 63.3 | 63.3 |
| Min stderr (3.3.2.1) | confidence | **82.7** | **79.9** | **80.5** | **79.7** | **77.7** |

Table 2: **VLMaps Replanning Subgoal Success Rates.** Similar to table 1, the baseline methods and 'Replan Strategy' column are explained in 4.1.1 and 3.3, respectively. The 'Selection Criteria' column describes the criteria for generating the high-confidence candidate set, as detailed in 3.4.

### 4.1.2 Evaluation Metric.

Given an object mask retrieved with the text query, we calculate its point cloud IoU with the GT mask corresponding to the query. We count it a success if the IoU between GT mask and the retrieved mask is greater than 0.25.

### 4.1.3 Results.

The best baseline entry from Table 1 is the "Random from top-2 confidence" method, which surpasses the naive maximum confidence baseline. This suggests that the map begins to provide biased confidence after the first failed retrieval.

Our experiments demonstrate that both single-view and multi-view methods can enhance replanning performance. Specifically, 20 out of 25 experiment entries outperformed the best baseline entry, highlighting the efficacy and robustness of replanning with uncertainty and multi-view consistency.

### 4.2 VLMaps.

To show the effectiveness of our method on the downstream task brought by the improvement of the object retrieval performance, we further conducted some experiments on object navigation tasks using VLMaps. Following the setup of [5], we randomly generate 10 scenes as well as some random poses for building the maps, and 10 object navigation tasks with some subtasks and subgoals for each scene. We use the Matterport3D dataset with HabitatSim simulator[13, 14, 12] for the agent to perceive and navigate. If the navigation fails, we then propose a new position of the specified class and check whether it is near the correct object. Due to the computational budget, we only select those methods with competitive results in the previous section.

### 4.2.1 Evaluation Metric.

Given a proposed point for replanning, we follow the settings of VLMaps which checks whether the distance from the point to the nearest specified object is less than 1 meter.

### 4.2.2 Results.

Table 2 showcases the effectiveness of various replanning strategies compared to the "Max confidence (highest score)" baseline. The "Max entropy" strategies demonstrate significant improvements over the baseline, reducing bias by considering entropy. The "Min stderr" strategy consistently outperforms all other methods, including the baseline, by focusing on minimizing the standard error in confidence predictions. This approach proves to be the most robust and effective, highlighting the importance of incorporating uncertainty and multi-view consistency in replanning.

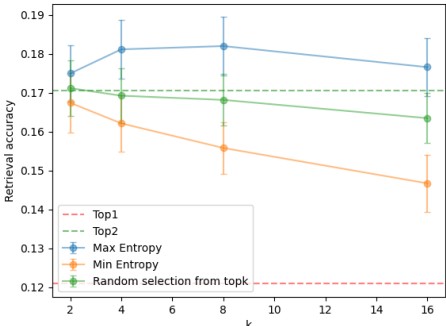

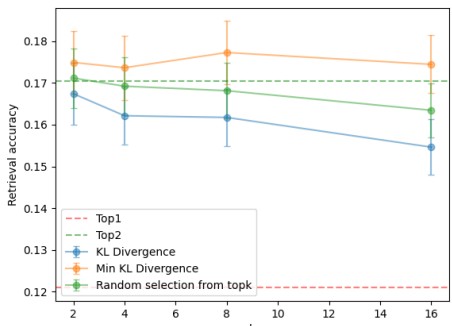

Figure 3: **Entropy vs. K.** If a model is biased, selecting low-entropy candidates from it might further reinforce the bias and ultimately degrade performance. This figure illustrates the performance of choosing minimum entropy targets from top-k confidence candidates.

Figure 4: **KL Divergence vs. K.** When the multi-view prediction is noisy, it's more likely to provide false information. This figure depicts the performance of selecting the least consistent candidate (i.e., maximum mean pairwise KL divergence) from top-k confidence candidates.

### 4.3 Ablation Studies: Different Choice on Choosing Maximal/Minimal Uncertainty.

To further support our assumptions, we conducted an ablation study with the OpenMask3D setting by examining different directions of the uncertainty metric. Results in figures 3 and 4 indicate that the variants consistently underperform the baselines.

## 5 Limitation

While our proposed CARe effectively cooperates with existing pre-explored semantic maps and navigation models in a training-free manner to achieve better performance, it may be limited by one major assumption: CARe assumes that the navigation model has consistent decision biases. This assumption holds true when working with a frozen model. However, if the navigation model is updated, these decision biases may be eliminated or changed, resulting in less significant performance improvements from CARe. While our proposed uncertainty measures are theoretically modality-agnostic and applicable to various 2D or 3D modalities, practical challenges such as sensitivity to channel scale in feature-based methods and the need for classifiers or alternative approaches in distribution-based methods should be considered to ensure their effectiveness. Additionally, because the structure of the semantic map may vary, CARe only uses the fixed semantic map for re-planning and does not further update the map with new information during the process. This research direction has the potential to continuously improve performance but is beyond the scope of this study. We will discuss and verify this direction in our future work.

## 6 Conclusion

**Summary:** We propose a novel method, Context-Aware Replanning (CARe), which accounts for unavoidable errors in Pre-explored Semantic Map and revises the plan. By leveraging uncertainty and multi-view consistency in Pre-explored Semantic Map, we replan the agent without additional human effort. We demonstrate the effectiveness and robustness of CARe by integrating it with two Pre-explored Semantic Map backbones, VLMaps [5] and OpenMask3D [3]. This integration consistently outperforms all baselines with various hyperparameters, achieving a peak success rate of 18.75% and a subgoal success rate of 82.7%.

**Future Work:** Given the portability of CARe, we are optimistic about its potential benefits for future robotic research and applications. By leveraging CARe, future research areas such as visual-and-language navigation (VLN) and applications like healthcare robots could facilitate replanning with reduced labeling costs. For example, a VLN system can leverage Pre-explored Semantic Map and enhance its success rate and Success weighted by Path Length (SPL) by replanning. The concept of CARe can also be adopted in the pre-exploring stage to enhance the quality of the Pre-explored Semantic Map by measuring with uncertainty and re-exploring.

**Acknowledgments**

This work was supported in part by National Science and Technology Council, Taiwan, under Grant NSTC 112-2634-F-002-006. We are grateful to MobileDrive and the National Center for High-performance Computing.

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

**Appendix**

# Part I

# Table of Contents

# A   More Method Details.

## A.1   Pseudocode of CARe.

To further enhance the reproducibility of our work, we present CARe's pseudocode in Algorithm 1. Additionally, we will open-source the code once our work is accepted.

## A.2   CARe with OpenMask3D .

**Method Overview:**   Following OpenMask3D [3], a transformer-based 3d instance segmentation model is used to propose 3d masks. After the masks are proposed, up to 5 views where the object is visible can be selected for calculating the mask feature.

**Main Augmentation:**   To enable multi-view uncertainty calculation, we augmented OpenMask3D not to take the average of the multi-view features, but to record features from all visible views for each object mask.

Let:

- $P$ be the scene point cloud.
- $I = \{I_1, I_2, \ldots, I_n\}$ be the set of posed RGB images.
- $M = \{m_1, m_2, \ldots, m_k\}$ be the set of class-agnostic 3D masks generated on the point cloud.
- $V_i$ be the set of views where object $m_i$ is visible.

For each visible object $m_i$: 1. Extract the semantic features using CLIP with cropped images from the corresponding views:

$$F_{ij} = \text{CLIP}(\text{crop}(I_j, m_i)), \quad \forall I_j \in V_i$$

where $F_{ij}$ is the feature vector for object $m_i$ from view $I_j$.

To handle multi-view features: - Instead of averaging the features, record all feature vectors for each object mask:

$$F_i = \{F_{ij} \mid I_j \in V_i\}$$

This recorded set of features $F_i$ for each object $m_i$ enables both single-view and multi-view uncertainty calculations as described in the previous sections.

**Algorithm 1** Pipeline of CARe
***

1: $S$ = SelectionMethod $\in$ {TopConfidence, TopPrediction}
2: $k$ = The boundary of ranking for the selection method
3: $U$ = UncertaintyMeasure $\in$ {Entropy, StandardError, KL}
4: $N$ = Number of all the points
5: $f$ = Feature dimensions
6: $m$ = Number of all possible classes
7: $O$ = Set of every point, shape $(N, f)$

8: **if** original plan fails **then**

9:     **if** $S$ == TopConfidence **then**                   ▷ Filter with selection method
10:         $C$ = score of $O$ from high to low, shape $(N, 1)$
11:         $O'$ = points with top $k$ highest scores, shape $(N', f)$
12:     **else if** $S$ == TopPrediction **then**
13:         $P$ = predicted class from high to low for each point in $O$, shape $(N, m)$
14:         $O'$ = points where their top $k$ predicted classes include the target class, shape $(N', f)$
15:     **end if**

16:     **if** $U$ == Entropy **then**             ▷ Choose a point with uncertainty measure
17:         $E$ = entropy of $O'$, shape $(N', 1)$
18:         $O^*$ = argmax(E), shape$(1, f)$
19:     **else if** $U$ == StandardError **then**
20:         $SE$ = standard error of $O'$, shape $(N', 1)$
21:         $O^*$ = argmin(SE), shape$(1, f)$
22:     **else if** $U$ == KL **then**
23:         $KL$ = KL divergence of $O'$, shape $(N', 1)$
24:         $O^*$ = argmin(KL), shape$(1, f)$
25:     **end if**

26:     **return** $O^*$
27: **end if**
***

**View Selection:** Given a 3d object mask, the 3d points in the mask are projected back to 2d for all posed RGB images in the scene. We then validate whether the object is visible in a view by checking if any of the points projected to 2d lies within the image. If there are more than 5 views where the object is visible, we rank the images by the number of object pixels and choose the top 5 views. This not only helps us manage the computation cost but also encourages the selection of views that are closer to the object, which might be helpful in filtering out far-away views that might not capture the object clearly.

**Feature Extraction:** Following the original OpenMask3D implementation, we used CLIP-ViT-L/14 for encoding images and texts. The 3d to 2d projection operation mentioned in the last paragraph has also allowed us to calculate the bounding-box of the object which we refer to as "object crops". For feature extraction, we encode the object crops with the CLIP visual encoder and save them all instead of taking the average of them. In the original OpenMask3D implementation, they also used multi-scale cropping for each object crop as a data augmentation. Since data augmentation is orthogonal to the direction of this work, we omitted this part for simplicity.

### A.3   CARe with VLMaps.

**Method Overview:** Following the original VLMaps [5], we first select 10 scenes and randomly generate several poses, which includes position and rotation, with their corresponding RGBD observations. Then, the image features generated by LSeg[10] model are projected to the global frame.

**Channel-average Feature Standard Error for VLMaps:** Let $C = \{c_1, c_2, \ldots, c_n\}$ be the set of candidates, and $V_i = \{v_1, v_2, \ldots, v_{m_i}\}$ be the set of views for candidate $c_i$. For a candidate $c_i$ with only a single view feature, we set the uncertainty score $U(c_i)$ to infinity.

Recall that for each candidate $c_i$ with multiple views, we compute the standard error $SE(c_i)$:

$$SE(c_i) = \frac{1}{d} \sum_{k=1}^{d} \frac{\sigma_k}{\sqrt{m_i}}$$

where $\sigma_k$ is the standard deviation of the $k$-th feature across all views, $d$ is the dimensionality of the feature vector, and $m_i$ is the number of views for candidate $c_i$.

And in the case of VLMaps, we incorporate the distance weighting [1, 5] information of each feature which is utilized when building the map, that is we replace the standard deviation and the sample size with the weighted version:

$$SE^w(c_i) = \frac{1}{d} \sum_{k=1}^{d} \frac{\sigma_k^w}{\sqrt{m_i^{eff}}}$$

where $\sigma_k^w$ is the weighted standard deviation and $m_i^{eff}$ is the effective sample size, calculated as follow:

$$m_i^{eff} = \frac{(\sum_{i=1}^{n} w_i)^2}{\sum_{i=1}^{n} w_i^2}$$

Finally, we choose the candidate $c^*$ with the lowest standard error.

**Map Generation:** As previously mentioned, we build the map by the method that is identical to VLMaps. However, we further save some metrics for each grid which are utilized in our work, such as entropy, standard error, and KL divergence. Additionally, to align with the method of feature fusion [1] adopted in VLMaps, we calculate the above metrics in their weighted version.

**Navigation and Planning:** In the navigation stage, VLMaps first generate a mask indicating the presence of a specific object class, and it then plans a path to the boundary of the nearest object. With the provided path, it further calculate the angle and distance between two subsequent halfway point, generating the low-level actions that is used in the HabitatSim [13, 14, 12].

**Evaluation and Re-proposing:** After all the actions are executed, we calculate the distance between the agent and the approximate boundary of the nearest object with the ground truth data provided by the simulator. Following the settings in VLMaps, we count it success when the distance is less than or equal to 1 meter. If it fails, we then generate a new proposal of where the object may be by our method CARe. Similarly, we calculate the distance between the new point and its nearest object, checking whether the distance is less than or equal to 1 meter.

## B   Experimental Details.

**Quantitative Results:** In the setting of VLMaps, we also conduct the KL divergence method as our uncertainty measure as shown in Table 3. However, some scenes with larger space or more data, may cause the calculation of pairwise KL divergence become quite computational intensive. Thus, we have to skip two larger scenes due to the limitation of memory size, which makes the result not comparable to others.

**Qualitative Results:** We provide more qualitative results on the anonymized project page[1], including the process of our proposed CARe solving an object navigation task. These qualitative results support our claims and make our work more convincing.

---

[1]https://carmaps.github.io/supplements/

| Replan Strategy | k=4 | k=8 | k=16 | k=40 | k=100 |
|---|---|---|---|---|---|
| No replan | 54.4 | 54.4 | 54.4 | 54.4 | 54.4 |
| Max confidence (highest score) | **67.0** | **67.0** | **67.0** | **67.0** | **67.0** |
| Random replan | 56.2 | 56.2 | 56.2 | 56.2 | 56.2 |
| Min KL from topk confidence | 82.7 | 83.1 | **85.2** | 84.5 | 82.7 |
| Min KL from topk category | 79.2 | 77.1 | 73.9 | 64.4 | 64.4 |

Table 3: VLMaps Replanning Subgoal Success Rates with KL divergence

## C  Supplementary Experiments and Analysis.

### C.1  Computational Complexity Analysis.

Our method incorporates a replanning phase when the initial retrieval attempt is unsuccessful. Such replanning phase includes the calculation of uncertainty measures, which introduces additional computational time compared to the baseline methods. To evaluate the computational overhead introduced by our approach, we performed a supplementary experiment and analysis.

| Replaning Strategy | entropy | stderr | pwKL |
|---|---|---|---|
| Time Complexity | $O(n)$ | $O(n)$ | $O(n^2)$ |
| Space Complexity | $O(n)$ | $O(n)$ | $O(n^2)$ |

Table 4: Complexity Analysis with Respect to Candidate Count $n$.

Table 4 presents a theoretical analysis for each replanning strategy. The computational complexity for the **entropy** and **stderr** measures increases linearly, whereas, for the **pwKL** measure, the complexity grows quadratically due to the need for pairwise computations between candidates. While the pairwise KL divergence method provides superior retrieval performance in the main experiments, this theoretical analysis suggests that it might suffer from larger computational costs when the candidate count $n$ is too large. In this case, users could consider using the **stderr** metric with linear complexity to measure multi-view consistency.

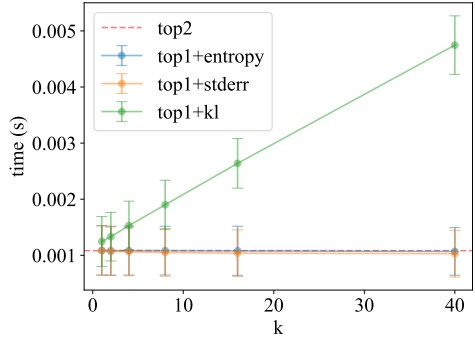

Figure 5: Latency for Top-k Confidence.

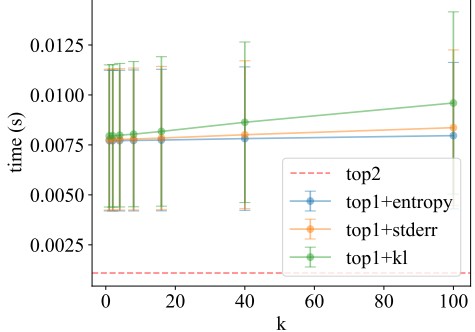

Figure 6: Latency for Top-k Category.

To further demonstrate the real-time applicability of our method, we measured the time requirements in the OpenMask3d experiment setting. We assumed that object features are precomputed and stored in the maps, with uncertainty measures such as entropy and KL divergence computed on the fly. Figures 5 and 6 present the results under different top-k strategies. For all uncertainty measures, both top-k selection criteria and all tested values of $k$, a retrieval attempt typically takes less than 15 milliseconds. Considering that real-world physical navigation involves seconds or minutes of path-following interactions after determining a destination, we believe that the millisecond-level computational overhead introduced by our method is negligible.

Note that in practice, the uncertainty measures can also be precomputed like the object features. In this case, the latency for our method could be further decreased. In latency-critical scenarios, users of our method can trade space for time by precomputing and storing the uncertainty measures.

## D   Limitations and Future Works.

While our proposed CARe effectively cooperates with existing pre-explored semantic maps and navigation models in a training-free manner to achieve better performance, it may be limited by one major assumption: CARe assumes that the navigation model has consistent decision biases. This assumption holds true when working with a frozen model. However, if the navigation model is updated, these decision biases may be eliminated or changed, resulting in less significant performance improvements from CARe. Additionally, because the structure of the semantic map may vary, CARe only uses the fixed semantic map for re-planning and does not further update the map with new information during the process. This research direction has the potential to continuously improve performance but is beyond the scope of this study. We will discuss and verify this direction in our future work.

