# OpenReview forum: "Context-Aware Replanning with Pre-Explored Semantic Map for Object Navigation"
_robot-learning.org/CoRL/2024/Conference — CoRL 2024_

### Official Review · Reviewer_f959 · 2024-07-19
**The idea is great and meaningful but the experiments need to be improved**

**Originality:** 3
**Technical Quality:** 3
**Clarity Of Presentation:** 2
**Potential Impact:** 3
**Recommendation:** 3
**Confidence:** 5

**Review:**

Strength:
The initiative of computing uncertainty in current open-vocabulary mapping is important and meaningful. The authors proposed to utilize the top-k candidates of the labels, and compute their probabilistic distribution, entropy, multi-view feature standard deviation, and average pairwise view KL-divergence of the distribution prediction. The metrics proposed are intuitive and make sense. In the overview, the results show valid improvement of performance when the metrics are applied to evaluating the meaningful goals to navigate to.

Weakness:
The major drawback of the paper is the clarity of the writing (method and experiment).
1. The metrics in the paper are well defined but further explanation of such definition would be appreciated. For example, SE(c_i), why do the authors think this metric can reflect the uncertainty? Same for D_{KL}(c_i) etc. It would be great to add one to two sentences to explain the meaning of the metrics.
2. At the beginning of the method part, it would be great to have an overall introduction of how the proposed metrics can be integrated into the pipeline of open-vocabulary mapping. Or cover what will be introduced in the subsections. Currently, each subsection of the method part is meaningful to some extent but there are no clear connections among them.
3. The experiments have a lot of rows but is not clear how they matched with the introduced methods (which row means what metric). It would be great if the authors can define those rows and the metrics they mean. At the same time, there are a lot of redundant descriptions like "top k confidence"/"top k category". The author can introduce a new column with the title "selection criterium" or "select with top-k", and in each row write either "confidence" or "category". Same for "std"/"stderr"/"pwKL"
4. The k seems to be a hyperparameters and the authors should bold the best performing entry for each column.
5. The conclusion is not clear.
6. The limitation of the methods are not described.

**Quality Of The Limitations Section:**

1

**Questions For Rebuttal:**

Address the bullet points in the review.

**Robotics Focus:**

3

**Summary Of Paper:**

The authors proposed several metrics to evaluate the uncertainty in current open-vocabulary mapping methods and use those metrics to help with open-vocabulary object navigation in explored environment

**Summary Of Recommendation:**

The idea and initiative of the paper are good but more clarity is required. If the authors can improve it according to some of the reviews, I would raise the rating to one level higher.

---

### Official Review · Reviewer_2uSj · 2024-07-25
**Context-Aware Replanning with Pre-Explored Semantic Map for Object Navigation**

**Originality:** 3
**Technical Quality:** 3
**Clarity Of Presentation:** 3
**Potential Impact:** 3
**Recommendation:** 2
**Confidence:** 5

**Review:**

This paper presents a novel approach to object navigation by integrating uncertainty estimation into the decision-making process. The main contributions of this work are:

    Originality: The approach of using confidence scores and multi-view consistency to evaluate map uncertainty is innovative and addresses a critical gap in current methodologies.
    Technical Quality: The proposed method is rigorously tested using state-of-the-art map backbones, VLMaps, and OpenMask3D. The experimental results demonstrate substantial improvements over existing methods.
    Clarity: The paper is well-structured, providing clear explanations of the theoretical background, methodology, and experimental results.
    Impact: This research has the potential to significantly enhance the accuracy and reliability of robotic object navigation in real-world applications.

However, several areas need improvement:

    Keywords: The keywords are not appropriately set, which indicates a potential oversight in ensuring the quality of the paper. The authors should carefully review the details of the manuscript.
    Abstract: It is unusual to include citation numbers in the abstract. The authors should remove these references.
    Novelty: While the use of VLMs for semantic mapping has been extensively studied recently, the approach of using confidence scores and uncertainty measurements predates VLM-based semantic mapping. Therefore, the novelty of this work appears limited.
    Experimental Details: Some experimental values are missing, such as those in Tables 1 and 2. A more comprehensive evaluation is necessary.
    Limitations: The paper does not discuss the limitations of the proposed method. It would benefit the readers to understand the constraints and potential issues. Specifically, the applicability of this method beyond the VLMaps and OpenMask3D backbones should be addressed. Semantic maps come in various forms (2D, 3D, object-centered, RGB-D maps, neural network embeddings), and it is unclear which types are compatible with the proposed method.
    Planning Discussion: Although the focus of this research is on replanning, the section on the proposed method primarily discusses uncertainty quantification, confidence selection, and map representation, with insufficient detail on the actual planning process.

**Quality Of The Limitations Section:**

1

**Questions For Rebuttal:**

Could you provide more details on the computational cost of the proposed method, particularly regarding its applicability in real-time applications?
Do you have plans to conduct additional experiments to validate the effectiveness of the proposed method under different environments and conditions? If so, please provide details.

**Robotics Focus:**

2

**Summary Of Paper:**

This paper introduces Context-Aware Replanning (CARe) for object navigation using pre-explored semantic maps constructed through visual language models (VLMs). Unlike traditional approaches that assume the accuracy of the map, CARe estimates map uncertainty through confidence scores and multi-view consistency, allowing for the revision of erroneous decisions without additional labels. The effectiveness of CARe is demonstrated using two modern map backbones, VLMaps and OpenMask3D, showing significant improvements in object navigation tasks.

**Summary Of Recommendation:**

This paper introduces a novel approach to object navigation by incorporating map uncertainty estimation. While the technical quality and potential impact are high, there are concerns regarding the novelty, completeness of experimental results, and the discussion of limitations. The authors should address these issues to strengthen the paper.

---

### Official Review · Reviewer_ojnQ · 2024-08-03
**Context-Aware Replanning (CARe)**

**Originality:** 2
**Technical Quality:** 3
**Clarity Of Presentation:** 4
**Potential Impact:** 2
**Recommendation:** 3
**Confidence:** 4

**Review:**

The work is intuitive and the authors clearly explain their methodology and the evaluation metrics used. The ablation studies further strengthen the validity of their approach by demonstrating the impact of different uncertainty measures. The results presented in the paper are convincing and support the claims made by the authors (the reviewer appreciates the supplementary materials provided).

Clarity: The paper is well-written and organized, making it easy to follow the authors' thought process. The introduction effectively motivates the problem and highlights the limitations of existing approaches. The related work section provides a good overview of the relevant literature. The methodology is described in detail, with clear explanations of the uncertainty measures and the map backbones used. The experimental setup and results are presented clearly, and the ablation studies provide further insights into the method's effectiveness.

Originality: The use of uncertainty estimation and multi-view consistency for replanning is a unique approach that addresses the limitations of existing methods that assume map accuracy. The modularity and portability of the proposed method to different map backbones further enhance its originality.

Significance: The research addresses a significant challenge in robotic navigation, where map inaccuracies can lead to task failures. The proposed method has the potential to improve the robustness and reliability of robotic systems in real-world environments where perception is imperfect. The modularity and portability of the method make it potentially applicable to a wide range of robotic applications beyond navigation.

**Quality Of The Limitations Section:**

1

**Questions For Rebuttal:**

1. The evaluation is limited to object navigation tasks in a simulated and relatively constrained environment. Has there been any attempts at a real-world implementation of a real robotic system?
2. The paper does not discuss the proposed method's computational complexity. It would be beneficial to analyse the computational overhead and its impact on real-time performance.
3. The paper could benefit from a more detailed discussion of the limitations of the proposed method and potential future directions for research.

**Robotics Focus:**

3

**Summary Of Paper:**

The paper presents a novel approach, Context-Aware Replanning (CARe), to address the challenge of inaccurate maps in robotic navigation tasks that utilise pre-explored semantic maps. The core idea is to incorporate uncertainty estimation and multi-view consistency into the decision-making process, allowing the robot to revise its plan when the initial plan fails due to map inaccuracies. The method is evaluated on object navigation tasks using two different map backbones, demonstrating improved performance compared to traditional approaches.

**Summary Of Recommendation:**

The work is interesting and addresses a relevant question within object navigation. It is a relatively small incremental work with a modest implementation in simulation.

---

### Author Rebuttal · Authors · 2024-08-12

We thank the reviewers and the area chair for their valuable comments. We appreciate the recognition of the novelty in leveraging uncertainty estimation and multi-view consistency for replanning (AC, ojnQ, 2uSj, f959), as well as the acknowledgment of the technical quality and performance of the proposed components (AC, ojnQ, 2uSj, f959). We are also grateful for the positive feedback on the impact on the robotics community and real-world applications (ojnQ, 2uSj), as well as the presentation (AC, ojnQ, 2uSj).

Now we address three main concerns raised by the reviewers . Please check the revised paper for details.

**Discussion about Limitations**
- The Limitations section was placed in the Appendix due to space constraints. We have added a paragraph in to the Conclusion section (line 318-330) to discuss these limitations such as the assumption of a biased grounding model, potential map updates, and data modalities not covered in our experiments for building maps. Please kindly check the revised paper for details.

**Clarification of Writing**
- We made the following clarifications:
  - **Experiment Tables:**
    1. **Filled Missing Values:** Duplicated values for baseline entries where $k$ is irrelevant.
    2. **Formatting Improvements:**
        - **Bolded** top entries in columns.
        - Highlighted top entries in rows.
        - *Italicized* best baseline entries for comparison.
    3. **Simplified Method Names:** Separated criteria selection into an independent column.
    4. **Added Descriptions and References:** Included for better context.
  - **Methods Section:**
    1. **Overview:** Added a brief introduction to the framework.
    2. **Metric Explanations:** Provided intuitive explanations for each uncertainty measure.

**Computational Complexity Analysis**
- We included a theoretical analysis and an empirical experiment in our OpenMask setting. (Appendix. C)

*Table A: Complexity Analysis with Respect to Candidate Count n.*
|Replanning Strategy|entropy|stderr|pwKL|
|-|-|-|-|
|**Time Complexity** |$$O(n)$$|$$O(n)$$|$$O(n^2)$$|
|**Space Complexity**|$$O(n)$$|$$O(n)$$|$$O(n^2)$$|

- In the empirical experiment, the latency for retrieval using our method is typically under 15ms, which we believe is acceptable for real-world scenarios.

---

- Please kindly check our revised main paper (with highlighted color) in the attached file and let us know if there are still any unresolved concerns.
- Thank you again for reviewing our paper.

---

### Decision · Program_Chairs · 2024-09-04

**Decision:**

Accept

**Comment:**

The paper addresses an interesting area of research. The authors have reasonably addressed the reviewers' concerns improving the quality of the paper. However, there is still room for improvement in terms of real-work experiments.